# Natural Vector of Avian *Haemoproteus asymmetricus* Parasite and Factors Altering the Spread of Infection

**DOI:** 10.3390/insects14120926

**Published:** 2023-12-05

**Authors:** Rita Žiegytė, Vaidas Palinauskas, Rasa Bernotienė

**Affiliations:** Nature Research Centre, Akademijos 2, 08412 Vilnius, Lithuania; vaidas.palinauskas@gamtc.lt

**Keywords:** *Culicoides*, *Haemoproteus*, sporozoites, haemosporidian infections, transmission, factors

## Abstract

**Simple Summary:**

Avian haemoproteids are prevalent parasites known to induce pathology or mortality in birds. *Culicoides* biting midges act as vectors for these parasites. Despite the annual arrival of various haemoproteids to Europe with migrant birds, not all these parasites undergo local transmission. The factors constraining the local spread of these pathogens remain incompletely understood. Our study investigates if the ecological isolation of birds from vectors, either temporally or spatially during the breeding period when transmission occurs, could lead to the cessation of transmission. Biting midges were systematically collected from two distinct habitats between May and September. A total of 1135 parous *Culicoides* females were identified and examined for the presence of *Haemoproteus* parasites using both microscopy and molecular tools. Notably, we report the first detection of *Haemoproteus asymmetricus* sporozoites in the salivary glands of *Culicoides festivipennis* females. The sporozoites of four *Haemoproteus* genetic lineages were also identified in *Culicoides segnis*, *C. festivipennis*, and *C. kibunensis*, further validating their status as potential *Haemoproteus* vectors. While the highest abundance of collected *Culicoides* females occurred in June, the peak prevalence of *Haemoproteus* parasites in biting midges was observed in July. Interestingly, the abundance of *Culicoides* was significantly greater in woodlands compared to reeds throughout the season.

**Abstract:**

Avian haemosporidians (Apicomplexa, Haemosporida) are widespread blood protists, often causing severe haemosporidiosis, pathology, or even mortality in their hosts. Migrant birds regularly bring various haemosporidian parasites from wintering grounds to European breeding areas. Some haemosporidian parasites are prevalent in breeding sites and complete their life cycles in temperate climate zones and can be transmitted, but others do not. The factors altering the spread of these haemosporidians are not fully understood. *Culicoides* biting midges (Diptera: Ceratopogonidae) play an important role in the transmission of worldwide distributed avian haemosporidian parasites belonging to the genus *Haemoproteus*, but this information is particularly scarce and insufficient. The key factors limiting the spread of these pathogens in temperate climate zones, which we suspect and aim to study, are the absence of susceptible vectors and the ecological isolation of birds from vectors during the breeding period when transmission occurs. The primary objective of this study was to evaluate how the habitats of biting midges and bird breeding sites influence parasite transmission while also seeking to expand our understanding of the natural vectors for these parasites. Biting midges were collected using UV traps on the Curonian Spit, Lithuania, in different habitats, such as woodland and reeds, from May to September. Parous *Culicoides* females were identified, dissected, and investigated for the presence of *Haemoproteus* parasites using both microscopy and PCR-based tools. Among the dissected 1135 parous *Culicoides* females, the sporozoites of *Haemoproteus asymmetricus* (genetic lineage hTUPHI01) have been detected for the first time in the salivary glands of *Culicoides festivipennis*. The sporozoites of four *Haemoproteus* lineages were detected in *Culicoides segnis*, *C. festivipennis*, and *Culicoides kibunensis* biting midges. PCR-based screening showed that the females of seven *Culicoides* species were naturally infected with *Haemoproteus* parasites. The DNA of the parasite of owls, *Haemoproteus syrnii* (hSTAL2), was detected for the first time in *Culicoides punctatus*. The highest abundance of collected *Culicoides* females was in June, but the highest prevalence of *Haemoproteus* parasites in biting midges was in July. The abundance of *Culicoides* was higher in the woodland compared with reeds during the season. The acquired findings indicate the varied abundance and diversity of biting midges throughout the season and across distinct habitats. This variability could potentially impact the transmission of *Haemoproteus* parasites among birds with diverse breeding site ecologies. These outcomes hold the potential to enhance our understanding of the epizootiology of *Haemoproteus* infections within temperate climatic zones.

## 1. Background

Outbreaks of new vector-borne haemosporidian infections have been frequently recorded during past decades, calling for research on wildlife pathogens to better understand their epizootiology and the main factors altering the spread of new diseases [1,2,3,4].

Migrant birds regularly bring various haemosporidian parasites from wintering grounds to European breeding areas and serve as reservoirs of potential disease outbreaks [5,6]. Some of these parasites can complete their life cycles in temperate climate zones and can be transmitted there, but others are transmitted only in wintering grounds [7,8,9,10,11]. The factors preventing the spread of these vector-borne infections are not studied sufficiently [11]. The primary factors that plausibly constrain the spread of these pathogens within temperate climatic regions are the lack of susceptible vectors and the ecological segregation of birds from vectors during their reproductive phase. The determination of these constraining factors holds significance due to ongoing climate change, which has caused ecological changes in Europe. Particularly in the southern regions, the expanding presence of hematophagous arthropods (vectors) in novel territories could potentially establish suitable environments for the emergence of hitherto unrecorded avian infections [3,10,12,13,14,15,16]. According to some predictions, an increase in the global temperature by 1 °C will co-occur with a two- to three-fold increase in avian malaria prevalence [17]. It alerts us to the significant changes in the epizootiological situation in temperate zones in the near future. Therefore, the main goal of our study is to investigate the epizootiology and transmission capabilities of *Haemoproteus* parasites, the factors limiting the spread of these pathogens in temperate climate zones, such as ecological isolation, and the success of transmission between birds and *Culicoides* females.

The abundance of vectors and the peculiarities of bird biology determine their contact with vectors and influence the prevalence of infection [7,18,19]. Much of the literature is focused on bird movement and the exchange of parasites between regions, but there is insufficient information about vector abundance, diversity, and activity, which would enhance our understanding of the ecology and evolution of host–parasite systems [20].

With the objective of ascertaining the ecological variables that impact the dissemination of parasites and uncovering of novel vector species, we collected female *Culicoides* specimens from two distinct habitats: a deciduous woodland primarily dominated by alders and a reed-dominated environment. Our investigation encompassed an evaluation of the population density, temporal patterns, and prevalence of avian haemosporidian infections, specifically within parous female specimens. The identification of new natural vectors is important because from more than 1400 Culicoides species described in the world [21], only 6 species have been identified as natural vectors of haemoproteids in Europe [22], and 10 more *Culicoides* species are known as haemoproteid vectors in the world [8,20].

The results of this study supplemented the list of vectors of haemoproteids in Europe and provided information about the factors influencing the spread of *Haemoproteus* parasites among birds. Studies focusing on avian *Haemoproteus* parasites and their vectors, as well as the dynamics of transmission, are still relatively limited. Consequently, these areas remain a significant priority within the field of haemosporidian research [20].

## 2. Materials and Methods

### 2.1. Study Site, Collection of Biting Midges 

Biting midges were collected in the Curonian Spit, located on the Baltic Sea, Lithuania, from May to September 2021. Four Biogents BG-Pro traps, hung at 1.5 m height, were used in two different habitats: two traps were hung in an old deciduous woodland dominated by alder (Figure 1a) and two traps were hung in reeds near the Curonian lagoon (Figure 1b). The distance between traps in the same habitat was on average 3.5 km, and the distance between traps in different habitats was on average 0.6 km. Additionally, one Onderstepoort 220 V UV trap was used in a woodland habitat at the same height to collect more parous *Culicoides* females for the detection of sporozoites in salivary glands, thereby targeting natural *Haemoproteus* vectors [22].

Insects were collected over a span of 17 nights, with a collection frequency of 2 nights in September, 3 nights in May, and 4 nights each in June, July, and August. The traps were turned on 1–2 h before sunset and turned off 2–3 h after sunrise. Insects were collected in a water container supplemented with a drop of liquid soap, as described by Bernotienė et al. [23]. The collected insects were transported to the laboratory of the Biological Station of the Nature Research Centre (Juodkrantė). Only parous *Culicoides* females were sorted according to Dyce [24] and identified according to Gutsevich [25], Glukhova et al. [26], and Mathieu et al. [27]. The material was studied under binocular stereoscopic microscopes Olympus SZ × 10 and Olympus B × 43 (Olympus Corporation, Tokyo, Japan), as described below.

### 2.2. Microscopic Examination of Preparations

The details of the dissection of parous biting midges were described by Valkiūnas [7] and Žiegytė et al. [22,28]. Briefly, each parous *Culicoides* female was individually dissected by removing the head and isolating the salivary glands from the breast into a drop of physiological solution on an objective slide. The content of the salivary glands was smeared, air-dried, fixed with absolute methanol, and stained with 4% Giemsa stain [22,28]. All residual parts of *Culicoides* female were fixed in 96% alcohol for PCR-based investigation. Representative preparations of sporozoites (49651NS–49652NS) were deposited in the Nature Research Centre, Vilnius, Lithuania. 

### 2.3. Molecular Analysis

The genetic material from the remains of each dissected *Culicoides* female was isolated utilizing the ammonium acetate DNA extraction method [29]. To identify avian haemosporidian parasites within the insects, we employed a nested PCR protocol described by [30,31] with outer primers HaemNFI/HaemNR3 and inner primers HAEMF/HAEMR2 to amplify the mitochondrial DNA cytochrome b (cyt b) gene segment of 479 bp of the parasite (*Haemoproteus* and *Plasmodium* spp.). To mitigate the risk of false positives, a negative control (H_2_O instead of target DNA) was included every 24 samples. 

To validate the morphological identification of *Culicoides* females that tested positive for haemosporidian parasites via PCR, a molecular examination of mitochondrial DNA cytochrome *c* oxidase subunit 1 (COI) was conducted using primers LCO1490 and HCO2198 [32]. The morphological identification consistently aligned with the PCR-based identification of biting midges, as the obtained sequences exhibited a 99–100% match with corresponding sequences from GenBank.

The DNA fragments from all PCR samples were visualized on 2% agarose gel using MidoriGreen dye (NIPPON Genetics Europe, Düren, Germany). Subsequently, all positive samples underwent sequencing using both forward and reverse primers, employing the Big-Dye^®^ Terminator v3.1 Cycle Sequencing Kit (Thermo Fisher Scientific, Vilnius, Lithuania) and an applied biosystems genetic analyzer 3500. The sequences were edited and aligned using BioEdit software version 7.2.5 [33]. Genetic lineages of the parasites were determined using the ‘Basic Local Alignment Search Tool’ (megablast algorithm) from NCBI BLAST (https://blast.ncbi.nlm.nih.gov/Blast.cgi, accessed on 20 August 2023), and their identification was cross-verified using the MalAvi database BLAST function (http://mbio-serv2.mbioekol.lu.se/Malavi, accessed on 20 August 2023).

### 2.4. Statistical Analysis

The statistical analysis was carried out using the Statistica 7 software package. The average values of the collected biting midges per night per one trap were calculated and provided with Standard error. The abundances of the midges collected in different habitats were compared using a *t*-test for dependent samples. A *p* value of 0.05 or less was considered significant.

## 3. Results

In total, 1900 *Culicoides* females were collected during May–September 2021 using four Biogents BG-Pro traps in four localities and two habitats. The peak abundance of *Culicoides* females occurred in June, with woodland areas exhibiting the highest count (reaching up to 273 females, with a mean of 111.1 ± 39 (SE) collected per night per trap). Lower abundances were observed in August in reeds (6.1 ± 4.0) and in both habitats in September (0.5 ± 0.5) (Figure 2). Among all biting midges, 67.7% were collected in woodland habitats, while only 32.3% were found in reeds. Despite utilizing the same collection methodology in both habitats, the abundance of collected *Culicoides* midges was statistically higher in woodland habitats compared to reeds (t = 2.41, *p* = 0.023).

In total, 857 of the *Culicoides* midges collected by using four traps were parous and dissected for the salivary gland preparations (Figure 2). Similar to the overall abundance of *Culicoides* females, the abundance of collected parous females was significantly higher in woodland habitats (72.7% of all collected parous biting midges) compared with the reeds (27.3%; t = 2.65, *p* = 0.013). The highest abundance of parous *Culicoides* females was recorded in June and July, with the highest relative abundance (proportion of parous females) ranging from 0 (reeds in May and September) to 72.9% (woodland in July).

During the investigation, we identified 10 different *Culicoides* species and the *C. obsoletus* complex. The dominant *Culicoides* species were *C. punctatus* (May and July), *C. obsoletus* complex (June), *C. kibunensis* (June), and *C. festivipennis* (from June to August). No statistical differences have been detected in the abundances of parous females for *C. punctatus* (*n* = 218), *C. kibunensis* (*n* = 105), and *C. obsoletus* (*n* = 83) in different habitats. However, differences in abundance among different habitats have been detected for *C. festivipennis* (*n* = 242, *p* = 0.046), *C. segnis* (*n* = 75, *p* = 0.027), and *C. impunctatus* (*n* = 45, *p* = 0.036) (Figure 3). *Culicoides pictipennis* was collected only in woodland (*n* = 33). Only a small number of parous females of *Culicoides reconditus*, *C. pallidicornis*, *C. nubeculosus,* and *C. grissescens* were collected during the investigation.

In total, 64 *Culicoides* females were detected to be PCR-positive for the presence of haemosporidian parasite DNA (Table 1). Thirty-four of these were collected in the woodland, exhibiting 10 genetic lineages of *Haemoproteus*, while 16 were obtained from reeds featuring 4 lineages (Figure 4, Table 1). A high prevalence of haemosporidian parasites in biting midges was determined in June (18 PCR-positive midges or 5.0% from all parous females), but the highest was determined in July (30 PCR-positive biting midges or 8.3%), while solely one PCR positive female biting midge was identified in both May and August (Figure 4).

An additional 278 parous *Culicoides* females were collected through the utilization of an Onderstepoort 220 V UV trap deployed within the woodland habitat. This approach was undertaken to enhance the number of studied natural vectors responsible for transmitting *Haemoproteus* parasites within their natural environment. Fourteen more PCR-positive females (5%) were detected from the material collected using the Onderstepoort trap (Table 1). 

June and July were the months with the highest number of *Culicoides* females with sporozoites detected in salivary glands (5 females collected in June and 9 females collected in July, Table 1). We detected sporozoites in the salivary glands of four PCR-negative *C. segnis* and one PCR-negative *C. kibunensis* females.

The DNA of *Haemoproteus majoris* (genetic lineages hCCF5, hPARUS1, hPHSIB1), *Haemoproteus palloris* (hWW1), *Haemoproteus asymmetricus* (hTUPHI01), *Haemoproteus minutus* (hTURDUS2), *Haemoproteus belopolskyi* (hHIICT1), *Haemoproteus tartakovskyi* (hHAWF1, hSISKIN1), and *Haemoproteus* sp. (hCCF4) was detected in *Culicoides* females (Table 1). The parasite of owls, *Haemoproteus syrnii* (hSTAL2), was detected for the first time in *C. punctatus*. Mixed *Haemoproteus* infection, detected by double chromatogram peaks, has been detected in *C. kibunensis*. The DNA of *Plasmodium matutinum* (genetic lineage pLINN1) was also detected in *C. festivipennis*, *C. impunctatus*, *C. punctatus*, and *C. obsoletus* biting midges. Detecting *Plasmodium* DNA in *Culicoides* females illustrates the possible abortive development of these parasites in non-competent vectors.

The DNA and sporozoites of *H. asymmetricus* (hTUPHI01) were detected for the first time in *C. festivipennis* biting midges’ salivary glands (Table 1, Figure 5a,b), showing that this *Culicoides* species can be considered as a natural vector of *H. asymmetricus*. Sporozoites originating from *H. majoris*, *H. asymmetricus*, *H. tartakovskyi*, and *H. minutus* (belonging to genetic lineages hPHSIB1, hTUPHI01, hHAWF1, hTURDUS2, respectively) were identified within biting midges of the species *C. segnis*, *C. festivipennis*, and *C. kibunensis* (Table 1).

## 4. Discussion

We determined that the highest abundance of *Culicoides* females occurs in June, with an average of 72 ± 44 *Culicoides* females collected per night using one trap. Conversely, the lowest abundances were recorded in August (9.2 ± 5.1) and September (0.9 ± 0.5) in the investigated area. This seasonal pattern, characterized by the highest abundance of collected biting midges in June, aligns with findings from prior investigations [34]. This consistent trend has also been noted by other authors conducting research in the Curonian spit [26,35,36,37]. Interestingly, the highest abundance of parous *Culicoides* females was recorded in July (Figure 2), not at the same time as the total *Culicoides* abundance (June), and the highest relative abundance (the proportion of parous females from all females) was the highest in July (56.5 ± 6.9% and up to 72.9% in woodland habitat). The lowest relative abundance of parous *Culicoides* females was detected in May (7.61 ± 3.3%) and September (9.38 ± 6.0%) when no parous females were collected in two out of four traps). Biting midge females live 2–4 weeks [26], so the highest proportion of parous females, which had already had at least one gonotrophic cycle, may appear later in comparison with the highest total abundance of *Culicoides* females. July had the highest number of *Culicoides* females with sporozoites detected in salivary glands (30 out of 64). This information is important for scientists elucidating the natural vectors of haemosporidian parasites. Our results show that July is the best month for this kind of investigation as it can be the month of the most active transmission of haemosporidian parasites of some species. 

Birds of many species in Lithuania have offspring in June during the highest activity of *Culicoides* biting midges, so the observed overlapping of vector activity and the appearance of birds’ offspring should favor the transmission of infections to juvenile birds. However, June might be the month when biting midges have their first blood meal and are free of parasites. For instance, there are no records of *H. nucleocondensus* local transmission, a parasite of great reed warbler (*Acrocephalus arundinaceus*) in juvenile birds in northern Europe, although locally abundant *Culicoides* species could serve as vector candidates for *H. nucleocondensus* transmission [38,39]. The maximum hatching and fledging period of great read warblers occurs in June and vector abundance, according to our investigations, is high at that time [39]. However, this situation may change with changes in the arrival times of migrant birds due to climate change [40] or the shifting of the time to earlier months of the biggest abundance of biting midges. We have observed variations in the population density of female *Culicoides* in two distinct habitats, with reeds being less conducive to these insects comparing to woodland. This factor might additionally contribute to the disruption of parasite transmission. For example, great read warblers breed in reeds, where the abundance of *Culicoides* females and especially parous females is lower compared with the woodland habitats. This discordance in habitats might prevent the transmission of some infections from adult birds to juveniles. Most of the biting midges—67.7%—have been collected in woodland and 32.3% have been collected in reeds; 34 PCR-positive biting midges were collected in the woodland and 16 were collected in reeds (Figure 4). It is known that *Culicoides* midges are more abundant in moist woodlands and grassy groves, which serve as suitable habitats for both adult *Culicoides* biting midges [26,41,42] and their larval development. This correlation may be attributed to the conducive conditions for the breeding of *Culicoides* larvae in such environments. According to certain studies [43], the biting midges of some species (e.g., *C. impunctatus*) typically fly an average distance of 75 m from their breeding areas. However, over several days, females of other *Culicoides* species can cover longer distances, reaching approximately 2 km [44]. Only one *C. impunctatus* female was collected in reeds, while 44 females were collected in the woodland (Figure 3) during our investigation. Reeds can be unattractive to biting midges because they are exposed to more wind and have less favorable microclimate for a tiny fly to hide in.

In our study, the dominant *Culicoides* species were the same in both habitats. These species (*C. punctatus*, *C. obsoletus* complex, *C. kibunensis*, *C. festivipennis)* are also known to be among the dominant *Culicoides* species at other localities in Lithuania and in other countries in Europe [22,34,42,45]. However, based on our data, the abundance of certain *Culicoides* species exhibited significant variation across different habitats (Figure 3). Specifically, known vectors of *Haemoproteus* such as *C. segnis*, *C. impunctatus* [22,36], and *C. pictipennis* (the latter only was found in woodland) demonstrated distinct habitat preferences. It is noteworthy that the prevalent parasites identified within *Culicoides* females align predominantly with those associated with the avian genera of forest habitats, such as *Turdus* (hTUPHI01, hTURDUS2), *Carduelis* (hSISKIN1), *Parus* (hPARUS1), or *Fringilla* (hCCF5). This observation implies that instances of detecting positive midges in reed habitats infected with the hPARUS1 parasite likely stem from movement originating in the nearby forests. This is substantiated by the necessity for these midges to have fed upon forest birds, like *Parus majoris*, to contract parasites from this specific lineage.

The DNA and sporozoites of *H. asymmetricus* (hTUPHI01) were detected for the first time in *C. festivipennis* biting midges’ salivary glands (Figure 5). This parasite was described by Valkiūnas et. al. [46] and was predominantly reported in song thrush *Turdus philomelos* and was closely related with *H. minutus* (hTURDUS2), which can be found in the black bird *Turdus merula* [47]. Previously, two vector species of *H. asymmetricus* (hTUPHI01) were reported: *C. kibunensis* [28] and *C. segnis* [48]. We also detected sporozoites of this parasite in *C. segnis* and *C. kibunensis* during our study. Moreover, our study shows that *C. festivipennis* is also a vector of this avian parasite, and this is a new natural vector of *Haemoproteus* parasites in Europe.

The sporozoites of *H. majoris*, *H. asymmetricus*, *H. tartakovskyi,* and *H. minutus* (hPHSIB1, hTUPHI01, hHAWF1, and hTURDUS2, respectively) were detected in *C. segnis*, *C. pictipennis,* and *C. kibunensis* biting midges. *Haemoproteus majoris* (hPHSIB1) sporozoites have already been detected in *C. segnis* [48] and our investigation can confirm this. The parasites of other genetic lineages of *H. majoris* can complete sporogony in *C. segnis* (hCCF5) [22] and *C. impunctatus* (hPARUS1) [49] females.

*Haemoproteus tartakovskyi* was known to be transmitted by *C. impunctatus* [36] and *C. segnis* (hHAWF1) [22], and in laboratory this parasite is known to form sporozoites in *C. nubeculosus* (hSISKIN1) [50]. We detected sporozoites of hHAWF1 in wild-caught *C. segnis* during this study. 

*Haemoproteus minutus* (hTURDUS2) sporozoites were detected and confirmed by PCR in wild-caught *C. impunctatus* [51], *C. kibunensis* [28], *C. segnis* [48], and in laboratory-reared *C. nubeculosus* [52]. Similarly, our study confirmed that this parasite can complete sporogony in a new vector *C. kibunensis*. It can also be mentioned that the females of some *Culicoides* species (*C. pictipennis*, *C. segnis*, *C. impunctatus*) which were caught that were significantly more abundant in woodland compared with reeds are known as vectors of many *Haemosporidian* parasites, as detected by [22,48].

In summary, our findings suggest that the factors influencing the disruption of parasite transmission in certain bird species may be linked to vector biology. This includes variations in the timing of vector activity compared to bird hatching and fledging, differences in vector abundance and species diversity in various habitats, and a mismatch between the habitats where some bird species usually nest. We enhanced the understanding of the *Culicoides* species’ role as a vector for *Haemoproteus* parasites, contributing novel insights into the principal *Culicoides* species responsible for transmitting haemoproteids in Northern Europe. 

## 5. Conclusions

The sporozoites of *Haemoproteus asymmetricus* (genetic lineage hTUPHI01) have been detected for the first time in the salivary glands of *Culicoides festivipennis*. The sporozoites of four *Haemoproteus* species were detected in *Culicoides segnis*, *C. festivipennis*, and *Culicoides kibunensis* biting midges. PCR-based screening showed that the females of seven *Culicoides* species were naturally infected with *Haemoproteus* parasites. The DNA of the parasite of owls, *Haemoproteus syrnii* (hSTAL2), was detected for the first time in *Culicoides punctatus*, which indicates that the biting midges of this species naturally take bloodmeal on owls.

The highest abundance of collected *Culicoides* females was in June, but the prevalence of *Haemoproteus* parasites in biting midges was the highest in July. The abundance of *Culicoides* was significantly higher in the woodland compared with reeds during the season. The acquired findings indicate the varied abundance and diversity of biting midges throughout the season and across distinct habitats. This variability could potentially impact the transmission of *Haemoproteus* parasites among birds with diverse breeding ecologies.

## Figures and Tables

**Figure 1 insects-14-00926-f001:**
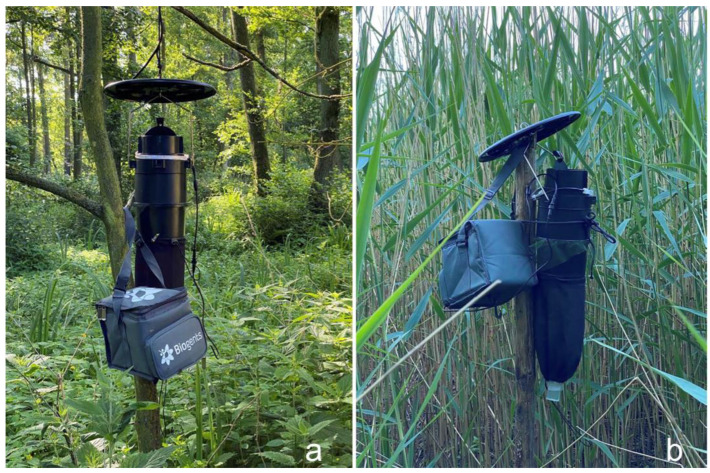
UV light traps used for *Culicoides* collection in woodland (**a**) and reeds (**b**).

**Figure 2 insects-14-00926-f002:**
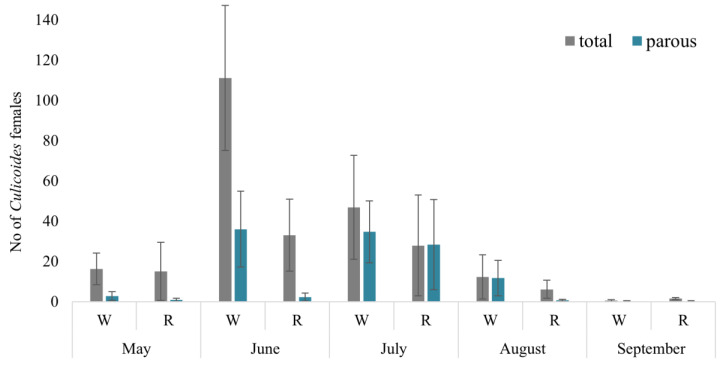
Mean number of *Culicoides* females collected per one night with one trap in different habitats. W—woodland, R—reeds.

**Figure 3 insects-14-00926-f003:**
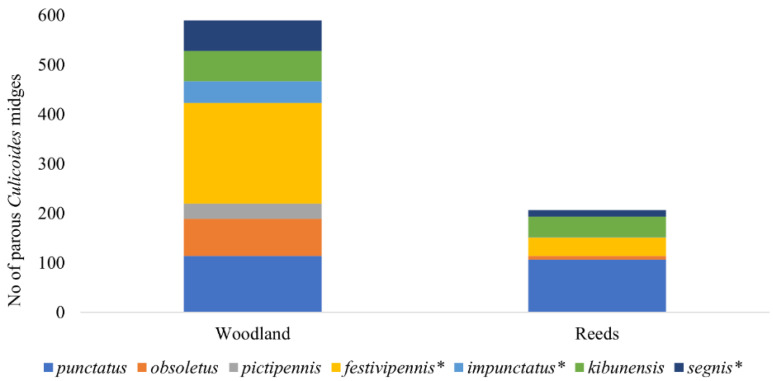
Number of parous *Culicoides* females collected in two habitats during the season. Asterisk indicates species whose abundances in different habitats differed significantly.

**Figure 4 insects-14-00926-f004:**
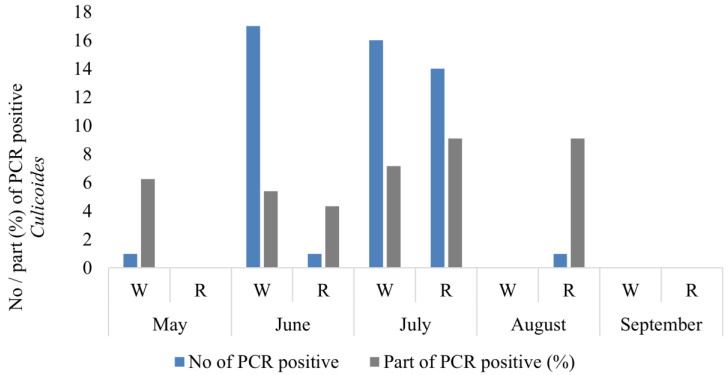
Number of PCR-positive *Culicoides* females collected during the warm season in two habitats. Part of PCR-positive (%) *Culicoides* females from all collected parous females. W—woodland, R—reeds.

**Figure 5 insects-14-00926-f005:**
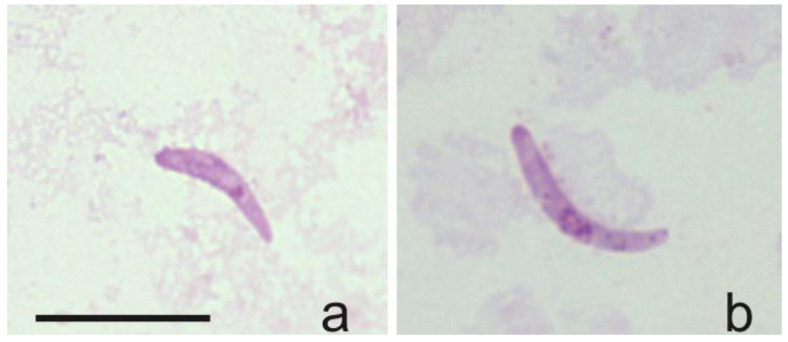
Sporozoites detected in salivary glands of two (**a**,**b**) individual *Culicoides festivipennis* biting midges. Genetic lineage was identified as *Haemoproteus asymmetricus* (hTUPHI01).

**Table 1 insects-14-00926-t001:** Parasites and genetic lineages detected in *Culicoides* females using PCR.

Parasite Species	Genetic Lineage	*Culicoides* Species	No. of Positive Females ^1^	No. of Positive Females ^2^
*Haemoproteus majoris*	hPARUS1	*C. punctatus*	11	1
		*C. obsoletus*	2	2
		*C. pictipennis*	2	
		*C. segnis*	1	
		*C. kibunensis*	1	1
	hCCF5	*C. segnis*	1	
	hPHSIB1	*C. segnis* *	1 *	
*Haemoproteus* sp.	hCCF4	*C. segnis*		1
		*C. reconditus*	1	
*Haemoproteus asymmetricus*	hTUPHI01	*C. festivipennis* *	4 *	
		*C. segnis* **	4 **	
		*C. kibunensis* *	2 *	
		*C. obsoletus*	1	1
		*C. punctatus*	1	
*Haemoproteus minutus*	hTURDUS2	*C. kibunensis* ***	1 *	2 **
*Haemoproteus palloris*	hWW1	*C. segnis*	1	
*Haemoproteus belopolskyi*	hHIICT1	*C. impunctatus*	1	
		*C. obsoletus*		1
*Haemoproteus tartakovskyi*	hHAWF1	*C. segnis* *	1 *	
		*C. festivipennis*	1	
	hSISKIN1	*C. impunctatus*	1	
*Haemoproteus syrnii*	hSTAL2	*C. punctatus*	1	
Mix infection		*C. kibunensis*		1
*Plasmodium matutinum*	pLINN1	*C. punctatus*	8	1
		*C. impunctatus*	2	1
		*C. obsoletus*	1	1
		*C. festivipennis*		1

^1^—UV traps from two habitats, ^2^—additional Onderstepoort 220 V UV trap. * *Culicoides* females with sporozoites detected in salivary glands. ** Cases with sporozoites detected in salivary glands of two *Culicoides* females. *** Cases with sporozoites detected in salivary glands of three *Culicoides* females.

## Data Availability

The data presented in this study are available upon email inquiry.

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
