# Peer review of "Natural Vector of Avian *Haemoproteus asymmetricus* Parasite and Factors Altering the Spread of Infection"

_insects, 2023, doi:10.3390/insects14120926_

Round 1

Reviewer 1 Report

Comments and Suggestions for Authors

This study aimed to explore the epizootiology of blood parasite infections in birds in the Curonian Spit. They found a significant difference in the Culicoides abundance between woodland and reed-dominated habitats and identified new vector species. There was also a lag between peak Culicoides abundance and peak parasite infection. The authors aptly discuss the importance of these findings in relation to overall bird health and make a very good point that changes in climate could negatively affect young birds if these abundance times were to shift. Overall, this was a nicely designed experiment and a well written paper. There are more specific comments and suggestions below.

L13: check with the journal’s requirements, but typically I see the order and family written with a colon. (Diptera: Ceratopogonidae)

L23: the information in the parenthesis here just seems off. Consider just stating the different habitats as a deciduous woodland and a lagoon. This is more descriptive of a habitat anyway, especially for the purposes of the abstract. This will change the wording of line 34.

L55-57: this sentence is a bit unclear. Do the authors mean “…due to ongoing climate change which has caused ecological changes in Europe.”

L65: add the word “and” before on “on the success”

L88: consider revising to “Biting midges were collected in the Curonian Spit, located on the Baltic Sea, Lithuania…”

L89-93: how far apart were the traps in the same habitats? How far apart were the two habitats?

L94: consider revising to “…glands, thereby targeting natural Haemoproteus vectors.”

L110: change to “…parous midges were dried, fixed…”

L113: this does not need to be a new paragraph.

L114-115: this information about the microscope is redundant with line 103 and can be deleted.

L131-132: after big dye, what machine was used to obtain the sequence data?

L145-152: I understand the information the authors are conveying here, but there are numerous grammatical errors throughout, and it will need revising.

L153: these are two of the worst shades of red and green for color blind people to differentiate. Please consider changing one of them to something else.

L156: delete “collected using four traps”. This information is redundant.

L160: change “reads” to “reeds”

L176: consider revising to “Thirty-four of these were collected in the woodlands…”

L185: change "No” to “Number” for clarity

L201-210: all the genus and species names need to be italicized in this paragraph.

L222: consider clarifying that this is for the Curonian Spit in the opening sentence.

L261-262: Dyce showed the average distance females disperse from a breeding site was 75m, not that they only stay with that range. He also was using stick cards with no attractant and there is much more evidence that host seeking females will travel 1-2 km in search of a blood meal. This does not diminish the authors’ findings of fewer C. impunctatus in the reed-dominated habitat and they make a good argument as to why this could be. Perhaps just reword the sentence to portray the citation more accurately or add the citations about host seeking at larger distances to strengthen the author’s case to explain the differences between habitats.

L272: consider changing “species” to “genera”

L275: change “moving” to “movement”

L299: change "rared” to “reared”

Is any of the genetic information obtained in this study, either the parasite or Culicoides sequences, available in any database such as GenBank? If not, I would encourage the authors to do so as this data is valuable to the scientific community.  

Comments on the Quality of English Language

The English is very good. There are some sections of the results to could use some refinement though.

Author Response

Rewiewer 1

This study aimed to explore the epizootiology of blood parasite infections in birds in the Curonian Spit. They found a significant difference in the Culicoides abundance between woodland and reed-dominated habitats and identified new vector species. There was also a lag between peak Culicoides abundance and peak parasite infection. The authors aptly discuss the importance of these findings in relation to overall bird health and make a very good point that changes in climate could negatively affect young birds if these abundance times were to shift. Overall, this was a nicely designed experiment and a well written paper. There are more specific comments and suggestions below.

We are very thankful for all comments and suggestions. We believe that these comments have improved the quality of the article.

L13: check with the journal’s requirements, but typically I see the order and family written with a colon. (Diptera: Ceratopogonidae)

Corrected as suggested (L 30)

L23: the information in the parenthesis here just seems off. Consider just stating the different habitats as a deciduous woodland and a lagoon. This is more descriptive of a habitat anyway, especially for the purposes of the abstract. This will change the wording of line 34.

Corrected as suggested (L 40)

L55-57: this sentence is a bit unclear. Do the authors mean “…due to ongoing climate change which has caused ecological changes in Europe.”

Corrected as suggested (L 73)

L65: add the word “and” before on “on the success”

Corrected as suggested (L 82)

L88: consider revising to “Biting midges were collected in the Curonian Spit, located on the Baltic Sea, Lithuania…”

Corrected as suggested (L 105)

L89-93: how far apart were the traps in the same habitats? How far apart were the two habitats?

Corrected as suggested (L 109-110)

L94: consider revising to “…glands, thereby targeting natural Haemoproteus vectors.”

Corrected as suggesteD (L 112)

L110: change to “…parous midges were dried, fixed…”

Thank You for this comment, we corrected similarly as suggested, because the sentence was corrected (L 130-131)

L113: this does not need to be a new paragraph.

Corrected (L 131)

L114-115: this information about the microscope is redundant with line 103 and can be deleted.

Deleted.

L131-132: after big dye, what machine was used to obtain the sequence data?

This information has been added (L 153)

L145-152: I understand the information the authors are conveying here, but there are numerous grammatical errors throughout, and it will need revising.

Corrected (L 166-173)

L153: these are two of the worst shades of red and green for color blind people to differentiate. Please consider changing one of them to something else.

Colors were changed and the figure was drawn as same axis of Figure 4, as suggested by another reviewer.

L156: delete “collected using four traps”. This information is redundant.

We used in total 5 traps: 4 traps were used in different habitats and 1 additional one was used to collect more parous Culicoides females for the detection of sporozoites in salivary glands, thereby targeting natural Haemoproteus vectors. We would like to leave this information, because in this paragraph we are talking only about biting midges collected with four traps (857 females), later we discuss biting midges collected with an additional trap (278 females). In total 1135 biting midges.

L160: change “reads” to “reeds”

Corrected (L 181)

L176: consider revising to “Thirty-four of these were collected in the woodlands…”

Corrected (L 199)

L185: change "No” to “Number” for clarity

Corrected (L 208)

L201-210: all the genus and species names need to be italicized in this paragraph.

Corrected (L 225-234)

L222: consider clarifying that this is for the Curonian Spit in the opening sentence.

Corrected. (L 249)

L261-262: Dyce showed the average distance females disperse from a breeding site was 75m, not that they only stay with that range. He also was using stick cards with no attractant and there is much more evidence that host seeking females will travel 1-2 km in search of a blood meal. This does not diminish the authors’ findings of fewer C. impunctatus in the reed-dominated habitat and they make a good argument as to why this could be. Perhaps just reword the sentence to portray the citation more accurately or add the citations about host seeking at larger distances to strengthen the author’s case to explain the differences between habitats.

Thank You for this comment. We supplemented the sentence and added information that biting midges can fly far more. We also added the citation about this [44] and L 291-292.

L272: consider changing “species” to “genera”

Corrected (L 305)

L275: change “moving” to “movement”

Corrected (L 308)

L299: change "rared” to “reared”

Corrected (L 332)

Is any of the genetic information obtained in this study, either the parasite or Culicoides sequences, available in any database such as GenBank? If not, I would encourage the authors to do so as this data is valuable to the scientific community.

All sequences of obtained genetic lineages (genetic lineage is a haplotype (479 bp) of cytochrome b of mtDNA) of Haemoproteus and Plasmodium parasites are already known from bird blood and sequences can be found in the GenBank or Malavi database (http://130.235.244.92/Malavi/), so we did not submitted known sequences into the GenBank.

Reviewer 2 Report

Comments and Suggestions for Authors

This manuscript describes a probable transmission cycle of Haemoproteus parasites among host birds and vector biting midges in Lithuania. Several genetic lineages of Haemoproteus were detected by PCR from the salivary glands of several Culicoides biting midges associated with clear observation of the sporozoites for the first time. The authors demonstrated the dynamics of infection of this parasite genus by showing the abundance of Culicoides during the transmission season and concluded that obtained results might provide significant knowledge to understand the natural occurrence of haemosporidian parasites and the spread of infection. This well analyzed study could contribute to know the variety of transmission of Haemoproteus parasites among host bird and can be acceptable as a scientific article after considering some modification for better expressions. So the reviewer recommends to revise this manuscript according to following comments.

Major concern:

Description of parasite detection should follow its order, namely, observed sporozoites can be genetically identified after confirmation of DNA sequence of amplified fragments. From this context, for example, the caption of Figure 5 can be modified as “Sporozoites detected in salivary glands of two individual Culicoides festivipennis biting midges. Genetic lineage was identified as Haemoproteus asymmetricus (hTUPHI01).”

Minor points:

Line 26-28: It can be modified as “Among dissected 1135 parous Culicoides females, sporozoites of Haemoproteus asymmetricus (genetic lineage hTUPHI01) have been detected for the first time in the salivary glands of C. festivipennis.”

Figure 2: This figure can be drawn as same axis of Figure 4.

Line 156: It can be modified as “857 of collected Culicoides midges by using four traps,”

Line 159: Please put “significantly” between “was” and “higher”.

Line 164 hereafter: Please consider to show how many Culicoides species was totally identified.

Table 1.: Layout of each names of parasite, lineage and Culicoides should be modified. Please put ‘centering’ off and left adjusted at each column.

Comments on the Quality of English Language

Minor editing of English language required.

Author Response

Reviewer 2

This manuscript describes a probable transmission cycle of Haemoproteus parasites among host birds and vector-biting midges in Lithuania. Several genetic lineages of Haemoproteus were detected by PCR from the salivary glands of several Culicoides biting midges associated with clear observation of the sporozoites for the first time. The authors demonstrated the dynamics of infection of this parasite genus by showing the abundance of Culicoides during the transmission season and concluded that obtained results might provide significant knowledge to understand the natural occurrence of haemosporidian parasites and the spread of infection. This well analyzed study could contribute to know the variety of transmission of Haemoproteus parasites among host bird and can be acceptable as a scientific article after considering some modification for better expressions. So the reviewer recommends to revise this manuscript according to following comments.

We are very thankful for all comments.

Major concern:

Description of parasite detection should follow its order, namely, observed sporozoites can be genetically identified after confirmation of DNA sequence of amplified fragments. From this context, for example, the caption of Figure 5 can be modified as “Sporozoites detected in salivary glands of two individual Culicoides festivipennis biting midges. Genetic lineage was identified as Haemoproteus asymmetricus (hTUPHI01).”

Thank You for this comment, this was corrected.

Minor points:

Line 26-28: It can be modified as “Among dissected 1135 parous Culicoides females, sporozoites of Haemoproteus asymmetricus (genetic lineage hTUPHI01) have been detected for the first time in the salivary glands of C. festivipennis.”

Corrected as suggested (L 43)

Figure 2: This figure can be drawn as same axis of Figure 4.

Corrected as suggested.

Line 156: It can be modified as “857 of collected Culicoides midges by using four traps,”

Corrected as suggested (L 177)

Line 159: Please put “significantly” between “was” and “higher”.

Corrected (179)

Line 164 hereafter: Please consider to show how many Culicoides species was totally identified.

This information has been added (L 184 and 191-193).

Table 1.: Layout of each names of parasite, lineage and Culicoides should be modified. Please put ‘centering’ off and left adjusted at each column.

Corrected.